# Paving the Way for Predicting the Progression of Cognitive Decline: The Potential Role of Machine Learning Algorithms in the Clinical Management of Neurodegenerative Disorders

**DOI:** 10.3390/jpm13091386

**Published:** 2023-09-15

**Authors:** Caterina Formica, Lilla Bonanno, Fabio Mauro Giambò, Giuseppa Maresca, Desiree Latella, Angela Marra, Fabio Cucinotta, Carmen Bonanno, Marco Lombardo, Orazio Tomarchio, Angelo Quartarone, Silvia Marino, Rocco Salvatore Calabrò, Viviana Lo Buono

**Affiliations:** 1IRCCS Centro Neurolesi “Bonino Pulejo”, 98124 Messina, Italy; katia.formica@irccsme.it (C.F.); lilla.bonanno@irccsme.it (L.B.); fabio.giambo@irccsme.it (F.M.G.); giusy.maresca@irccsme.it (G.M.); angela.marra@irccsme.it (A.M.); fabio.cucinotta@irccsme.it (F.C.); carmen.bonanno@irccsme.it (C.B.); angelo.quartarone@irccsme.it (A.Q.); silvia.marino@irccsme.it (S.M.); roccos.calabro@irccsme.it (R.S.C.); viviana.lobuono@irccsme.it (V.L.B.); 2Behavior Labs srl, 95030 Catania, Italy; marcolomby77@gmail.com; 3Department of Electrical Engineering, Electronics and Computer Science, University of Catania, 95131 Catania, Italy; orazio.tomarchio@unict.it

**Keywords:** artificial intelligence, machine learning, mild cognitive impairment, Alzheimer’s disease

## Abstract

Alzheimer’s disease (AD) is the most common form of neurodegenerative disorder. The prodromal phase of AD is mild cognitive impairment (MCI). The capacity to predict the transitional phase from MCI to AD represents a challenge for the scientific community. The adoption of artificial intelligence (AI) is useful for diagnostic, predictive analysis starting from the clinical epidemiology of neurodegenerative disorders. We propose a Machine Learning Model (MLM) where the algorithms were trained on a set of neuropsychological, neurophysiological, and clinical data to predict the diagnosis of cognitive decline in both MCI and AD patients. Methods: We built a dataset with clinical and neuropsychological data of 4848 patients, of which 2156 had a diagnosis of AD, and 2684 of MCI, for the Machine Learning Model, and 60 patients were enrolled for the test dataset. We trained an ML algorithm using RoboMate software based on the training dataset, and then calculated its accuracy using the test dataset. Results: The Receiver Operating Characteristic (ROC) analysis revealed that diagnostic accuracy was 86%, with an appropriate cutoff value of 1.5; sensitivity was 72%; and specificity reached a value of 91% for clinical data prediction with MMSE. Conclusion: This method may support clinicians to provide a second opinion concerning high prognostic power in the progression of cognitive impairment. The MLM used in this study is based on big data that were confirmed in enrolled patients and given a credibility about the presence of determinant risk factors also supported by a cognitive test score.

## 1. Introduction

Alzheimer’s disease (AD) is the most common form of neurodegenerative disorder, and it is characterized by a progressive loss of cognitive function, including memory [1,2]. In 2023, 6.7 million people 65 years old and older are affected by AD. This number could grow to 13.8 million by 2060, barring the development of medical breakthroughs to prevent, slow, or cure AD dementia [3]. This high prevalence has a significant impact on healthcare systems, not only in terms of economics, but also in terms of social burden. There is no specific pharmacological treatment to cure AD, but different therapies may alleviate the symptoms [4]. To date, the diagnosis is based on the collection of clinical data, instrumental examination, and neuropsychological assessment. In particular, the most accurate neuroimaging examinations to study the multisystemic degeneration of the brain (i.e., Positron Emission Tomography—PET—and functional Magnetic Resonance Imaging—fMRI) can individuate AD progression already in the early stage of the disease, thanks to specific biomarkers, such as tau protein fixation, amyloid markers, and presence of grey matter atrophy. The individuations of biomarkers, the use of a genetic approach, and the detection of neuropsychological and clinical signals could aid clinicians in the development of new treatments, monitoring the effectiveness of current and novel treatments, and reducing the time needed for these developments to avoid the misdiagnosis rate [5]. The prodromal phase of AD is termed mild cognitive impairment (MCI) [6,7]. According to the criteria of DSM-V, this disorder can be defined as one of the “mild neurocognitive disorders”, and it is characterized by the presence of one or more specific cognitive deficits that do not interfere in daily living activities [8]. The progression of AD from brain changes that cause memory problems, physical disability, and difficulties in performing daily living activities is called the AD continuum. This continuum includes three main phases: preclinical Alzheimer’s disease, mild cognitive impairment (MCI) due to Alzheimer’s disease, and dementia due to Alzheimer’s disease, also called Alzheimer’s dementia [9,10,11]. To our knowledge, the Alzheimer’s disease continuum starts with preclinical Alzheimer’s disease (no symptoms) and ends with severe Alzheimer’s dementia (severe symptoms). The continuum individually changes based on age, gender, and other environmental and biological factors. In a prospective of differential diagnosis, patients with MCI due to AD showed typical AD biomarkers and symptoms such as memory, language, and thinking deficits. These cognitive deficits may be known to relatives and caregivers, but do not interfere with individuals’ daily living activities. Cognitive decline occurs consequent to cerebral atrophy [12,13]. AD’s prodromal phase is MCI. About 15% of MCI patients develop dementia after two years, [14] while one third develop AD within five years [15]. However, some individuals with MCI do not have additional cognitive decline or revert to pre-disease cognition. A systematic review and meta-analysis of population-based studies reported a reversion rate of 26% [16,17]. To date, a challenge for the scientific community is to identify which MCI patients are more likely to develop dementia. The capacity to individuate the transitional phase from MCI stage (where symptoms are not necessarily indicative of an AD progression) becomes a condition of interest in the scientific and clinical community. For this reason, it is crucial to correlate the most evident clinical data and neuropsychological, and neurophysiological markers to allow the clinician to predict the progression of neurodegenerative disorders [18,19]. Although biological factors and family history cannot be changed, some risk factors could be modified to reduce the risk of cognitive impairment and dementia. Examples of modifiable risk factors are physical activity, smoking, education, mental activity, hypertension, diabetes, and diet. The recent discovery that AD can be diagnosed 20 years or more before the onset of symptoms suggests that there is a substantial window of time to intervene in the progression of the disease. Scientific advances in the identification of biomarkers for Alzheimer’s, such as beta-amyloid accumulation in the brain, enable earlier addressing of modifiable risk factors that may slow or delay cognitive decline. The identification and management of modifiable risk factors seems to be a good starting point in the absence of effective pharmacological therapies to treat dementia. One of the most studied methods in scientific research to optimize the control of numerous factors that have an incidence in the field of neurodegenerative pathologies is artificial intelligence (AI). The use of AI in medicine is an issue of great interest. The adoption of this technique has become valid for diagnostic and predictive analysis starting from the clinical epidemiology of a specific pathology. To date, the application fields of AI are microscopic pathologic images, metabolic disorders, and radiologic images for a deep learning of brain networks [20,21,22,23]. The development of algorithms to predict the course of a disease and make the proper diagnosis requires a huge dataset referred to “big data”. This is an essential phrase to describe the disease spectrum with the major representative characteristics to develop and verify the Predictive and Diagnostic Model of Machine Learning (ML). The algorithm should include routine clinical data used in clinical practice for a high predictive capacity. Considering the evidence available from the literature about sociodemographic characteristics and clinical risks (such as cardiovascular condition, metabolic syndrome, and atrophy of the brain), it may be useful to create a predictor algorithm to better manage neurogenerative disorders.

So, MLM could support the automatic screening phase with a high specificity for the prediction of cognitive decline; AI helps clinicians to identify cognitive impairment more accurately with a test, such as the MMSE, that has actual applicability in clinical practice and represents an early and timely way to obtain an accurate diagnosis in common clinical practice.

In this study, we propose an MLM where algorithms were trained on a set of neuropsychological, neurophysiological, and clinical data to predict the evolution of cognitive decline in MCI and AD patients.

## 2. Materials and Methods

We collected data from 4848 patients, of which 2156 had a diagnosis of AD and 2684 of MCI (see Figure 1). Data were obtained from the Dementia Outpatient Clinic of the IRCCS Centro Neurolesi, also known as the “Bonino Pulejo” Center, for the building of the MLM, and 60 patients were assessed to investigate the power and effectiveness of the prediction model. The enrollment took place from September 2022 to September 2023. Also, these patients were recruited to our institute from the Dementia Outpatient Clinic. We included sociodemographic variables, such as sex, age, and education, as well as clinical data, such as heredity, hypertension, diabetes, dyslipidemia, cardiovascular disease, atrial fibrillation, carotid stenosis, and smoking. Clinical scales (used to investigate the impact of dementia on activity and quality of life) and neuropsychological tests (to investigate the level of the cognitive impairment) were also administered. A total of 16 variables were then considered.

### 2.1. Clinical Scales

The Clinical Dementia Rating (CDR) [24] is a 5-point scale (0 = none, 0.5 = very mild, 1 = mild, 2 = moderate, 3 = severe) used to characterize six domains of cognitive and functional performance with dementia: Memory, Orientation, Judgment & Problem Solving, Community Affairs, Home & Hobbies, and Personal Care. Geriatric Depression Scale (GDS) is a 30-item yes-no self-report assessment used to identify depression in the elderly [25]. Activities of daily living (ADL) describe activities essential for self-care: bathing, dressing, and feeding (range 0–6). Instrumental activities of daily living (IADL) describe activities necessary for adaptation to the environment and community life such as shopping, cooking, and transportation, and are more cognitive-influenced (range 0–8 for female, 0–5 for male) [26].

### 2.2. Neuropsychological Test

The Mini Mental State Examination (MMSE) is the most commonly used assessment of mental state for elderly people. Cognitive areas indicated are spatial and temporal orientation, attention, memory, denomination, language, and execution of verbal commands [27]. The raw MMSE scores were categorized into 5 categories (0–4), adjusted for age and education (Table 1). A score of 0 represents the maximum severity of cognitive impairment, while 4 represents unimpaired cognitive level. This value was called Cognitive Status (CS).

### 2.3. Machine Learning Model (MLM)

The ML method assumes that computer systems can learn from data. The purpose of this technique is to provide software with the skill to learn from the data collected. ML was the most appropriate technology for the “Therapeutic Robot and Artificial Intelligence in experimental Therapy” project (T.R.A.I.T.), whose goal is also to perform predictions on patients with MCI. The aim is to suggest the best rehabilitative program according to the individual patient’s objective by identifying the patient’s cognitive impairment. A predictive statistical model (based on ML techniques) has been developed and indicated the presence or absence of the patient’s cognitive impairment, identified using the MMSE test scores. In MLM, classification is the process of predicting one or more classes on given data sets. The classification predictive model has the task of approximating a mapping function (f) from input variables (X) to discrete output variables (y). Classification belongs to the category of supervised learning in which, in addition to the target (data toward which the prediction is to be made), input data useful for constructing the relationship between them and the value to be predicted are also provided. The predictive function, starting from a large dataset, builds statistical relationships to indicate the probability that a patient may have cognitive decline and how this may evolve in the future by taking determinant clinical and sociodemographic information. The model is also able to highlight the role of risk factors and their impact on the patient’s overall cognitive level. The training of the predictive model starts with the construction of a defined dataset, which represents the model that is allowed to operate on new data and make predictions about the trend of these data. The system has a predictive function related to the evolution of the patient’s clinical view, obtained by matching the clinical data provided previously and the result of neuropsychological test scores. This method provided rehabilitative planning designed specifically for the patient. Two different prediction models have been implemented, and these models satisfy two important aspects in assessing a patient’s CS: i) the predictive training model, which aims to predict the patient’s CS considering the dataset collected, and (ii) the predictive model after assessment, which aims to evaluate the effectiveness of the predictive training model.

### 2.4. Machine Learning Context

-Systems: Windows, Linux, and iOS/iPadOs.-Software: RoboMate, Choreographe 2.5.10 (IDE), Photoshop/GIMP, Audiacity, and video editing software.-Programming: Python, HTML/CSS, Javascript, JSON, and NAOqi.

The nature of the RoboMate platform and Apple’s Neural Engine found in iPads is to have real-time predictive results, differently from a centralized approach where models are available through Web Services, with possible consequences of slowdowns and latencies due to the network or a high number of requests from remote devices. The first dataset was created manually through a procedure of collecting data from different sources. Data collection was realized through the RoboMate App, where the operator entered the clinical and sociodemographic data for each patient, including the score of the cognitive performance. The data were converted to CVS format and then the new revision of the two pre-assessment and post-assessment models was generated. RoboMate is a Learning Management System (LMS) software/platform, aimed at training people with neurocognitive disabilities. This employs a combination of training sessions, based on interaction through tablets and humanoid robots. The software architecture is robot-centric; the exercise is performed primarily by a robot, and the tablet displays the data that the robot hold in memory. RoboMate is presented as an SaaS (Software as a Service) Cloud platform. The app on iPad is a native iOS/iPadOS software, not available on the App Store but distributed ad hoc. The Choreographe project is created according to a Python block logic and Timeline blocks, each block defining one or more input parameters that are processed within the block and providing one or more outputs.

## 3. Statistical Analysis

### Diagnostic Accuracy

ROC analysis was performed to calculate the diagnostic accuracy Area Under the ROC curve (AUC) of system (null hypothesis: AUC = 0.5), with an appropriate cutoff. We compared the obtained results with clinical data prediction, and clinical data prediction with the MMSE (golden test). DeLong’s test used two correlated ROC curves. Sensitivity, specificity, negative and positive predictive values, and positive and negative likelihood ratios at designated cutoff levels with their 95% confidence interval (CI) were evaluated. A P-value less than 0.05 (two-sided) was considered to indicate statistical significances. The analysis was performed using the software R 4.2. A hypothesis test of interest is whether the clinical data prediction score and clinical data prediction with MMSE discriminates between the two groups. This problem can be reformulated in terms of AUC. The AUC is equal to 0.57 (95%CI: 0.40–0.72), a value for which the marker would not be very accurate in the clinical data prediction score, while the AUC is 0.86 (95%CI: 0.72–0.95), a value for which the marker would be moderately accurate in the clinical data prediction with MMSE score, according to the classification of Swets [28].

## 4. Results

The proposed method was applied to 60 patients. We studied 30 patients who were classified with severe cognitive impairment and 30 patients classified with moderate cognitive impairment when compared using the MMSE (golden test). DeLong’s test shows that the performance of clinical data prediction with MMSE score exceeded that of the clinical data prediction with difference between areas of 0.30 (95%CI: 0.12–0.47; *p* = 0.001). The optimal threshold value is the point with a shorter distance [29]. In this case, the output of the statistical tests shows that, for the Youden index, the optimal cutoff value is k = for each test. In particular, with k = 1.5, we found a specificity of 0.73 (39.1–94.0) and sensitivity of 0.45 (26.4–64.3) for clinical data prediction, and a specificity of 0.90 (58.7–99.8) and sensitivity of 0.72 (52.8–87.3) for clinical data prediction with MMSE. The ROC analysis revealed a highly significant AUC difference from 0.5 (null hypothesis) in the discrimination between patients; the diagnostic accuracy was 86%, with an appropriate cutoff value of 1.5; sensitivity was 72%; and specificity reached a value of 91% (see Figure 2).

## 5. Discussion

This study aimed to demonstrate the reliability of MLM trained on both neuropsychological measures and clinical data for performing a prediction of cognitive decline in MCI and AD patients. It seems that clinical risk factors, supported by neuropsychological measures, could lead to a successful automatic prediction about prodromal cognitive impairment in MCI patients. Moreover, clinical risk factors (such as hypertension, diabetes, dyslipidemia, and cardiovascular disease), supported by neuropsychological measures, can lead to successful automatic prediction of prodromal cognitive impairment in patients with MCI. MLM achieved a higher level of accuracy in clinical risk factors data combined with MMSE scores, with respect only to clinical risk factors.

The high interest in this neurodegenerative pathology could be due to the lack of pharmacological treatments that can slow down the development of the disease. Different studies have supported the use of MLM based on neuroimaging-related biomarkers for the differential diagnosis of AD, and to better understand the pathophysiology of neurodegenerative disorders [30,31,32]. A review by Salvatore et al. [30] demonstrates that an MLM based on a neuroimaging approach could classify patients who will or will not develop AD. In particular, the authors assessed the relevance of each brain voxel to the classification analysis, and identified regions involved in the pathophysiologic mechanisms of AD to distinguish clinically and cognitively compatible MCI patients who will progress to AD from those who will not. Other studies used neurophysiological data for the classification and potential evolution of neurodegenerative disease [33,34,35], demonstrating that an advanced neuroimaging and neurophysiological approach based on MLM was more accurate to classify patients who convert to AD or not, and to study brain regions involved in the pathophysiology of AD. New frontiers have considered the neuropsychological data for the construction of MLM [36,37,38]. In particular, Grassi et al. [37] examined sociodemographic data, clinical risk factors, brain atrophy, and neuropsychological test scores to develop an algorithm for 3-year prediction of conversion to AD from MCI. Unlike our study, the authors used more specific tests to investigate cognitive deficit. Certainly, the use of such domain-specific tests to build MLM provides more detailed data about the patient’s cognitive decline, compared to tests that evaluate only the global cognitive level of the patients. Youn’s study [38] provided evidence about the utility of an algorithm for the prediction of cognitive decline based on MMSE scores, with the objective to distinguish cognitively unimpaired and cognitively impaired patients. In our study, we used a similar method that could be effective to screen cognitive impairment in MCI patients based on MMSE score and clinical variables to predict cognitive impairment. Our model provided a sensitivity of 72% and specificity of 91% for clinical data with MMSE scores. These results represented a novel aspect to this literature study [38] that allows a lower specificity and insufficient accuracy to reach the level that is typically expected in MLM. Another difference with the model proposed by Youn’s study was that clinical variables were used only to screen MCI patients; in our work, clinical data represented the variables without which we would not have achieved such a high predictive specificity (91%). Two recent review studies [39,40] on the development of MLM in neurodegenerative disease showed that MLM and clinical data create new research opportunities, but the application of the training predictive model, combined with healthcare, is a challenge to overcome. It could be useful to develop an integrated approach for the implementation of clinical prediction and classification algorithms. This combined approach should connect the skills of medical professionals and researchers. However, if we consider the importance of early diagnosis and immediate intervention in the field of neurodegenerative diseases, the timeliness of the technique turns out to be important. Our findings suggest that MLM could support the automatic screening phase with more than 72% sensitivity for both groups of patients with moderate and severe cognitive impairment, and a specificity of 91% for the prediction of cognitive decline. Then, we may assume that our method may support the clinicians in more accurately identifying the progression of cognitive impairment. The MLM used in this study is based on big data that are confirmed in enrolled patients and given a credibility about the presence of determinant risk factors also supported by a cognitive tests score.

The use of the MMSE and only some of the clinical scales to assess behavior and activity of daily living could be considered a limitation of our study. The MMSE had an actual applicability in clinical practice; indeed, even if the tool alone can give approximate information, it can obtain an accuracy of 91% in identifying the prediction of the course of cognitive decline when combined with the clinical and sociodemographic data, as demonstrated in our study. For this reason, the main features of the test (easy and rapid to administer, short training to use the tool, etc.) could represent a strength. In fact, the greatest challenge is to have an early and prompt way to achieve an accurate prognosis through the introduction of AI techniques in common clinical practice. To this end, the MMSE could be an optimal solution.

## 6. Limitations

Using only MMSE scores, a clinical scale for the assessment of depression and ADL and IADL for activities of daily living, may represent a limitation of our study, compared to the use of a more complete neuropsychological battery. Moreover, the sample of 60 recruited patients should be enlarged and a follow-up evaluation should be carried out as further confirmation of the predictive result proposed by the MLM regarding the patient’s cognitive decline over time.

## 7. Conclusions

In conclusion, our MLM, using a rapid and easy-to-use clinical scale (i.e., the MMSE) could be considered a valuable tool in detecting cognitive decline and the progression to more severe forms of dementia. In particular, this model demonstrated that only MMSE scores associated with clinical data provided to the clinicians result in a specificity of 91% with respect to the cognitive decline of MCI patients. This finding highlighted the importance of AI use in this field and also demonstrated the actual applicability of the method. Further studies, using both neuroimaging and neurophysiological approaches, are needed to confirm the potential of ML in the early diagnosis of AD, in order to implement all treatments that can at least slow the progression of the disease.

## Figures and Tables

**Figure 1 jpm-13-01386-f001:**
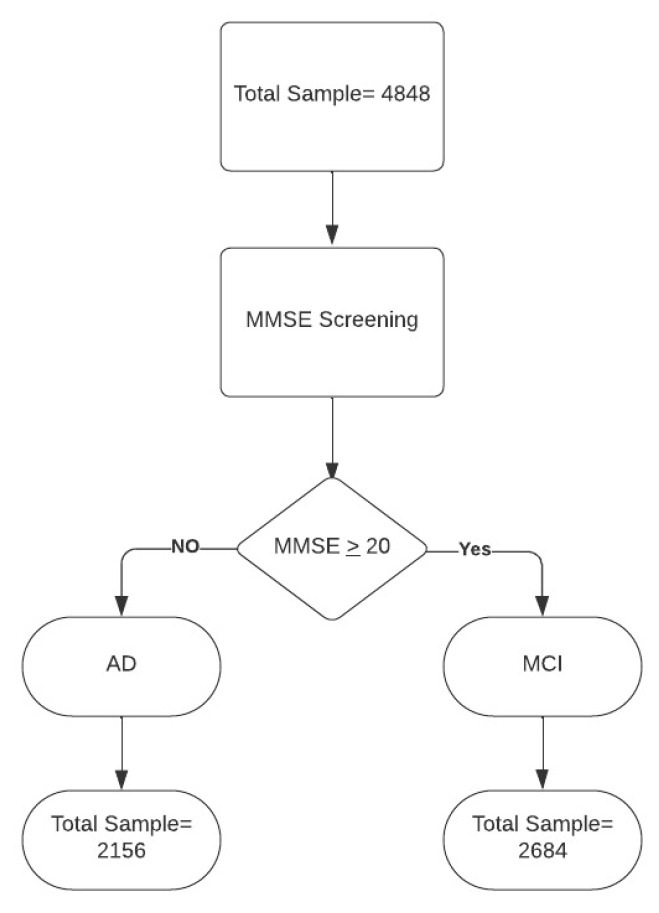
Data flow. Legend: MMSE = Mini Mental State Examination; AD = Alzheimer’s disease; MCI = mild cognitive impairment.

**Figure 2 jpm-13-01386-f002:**
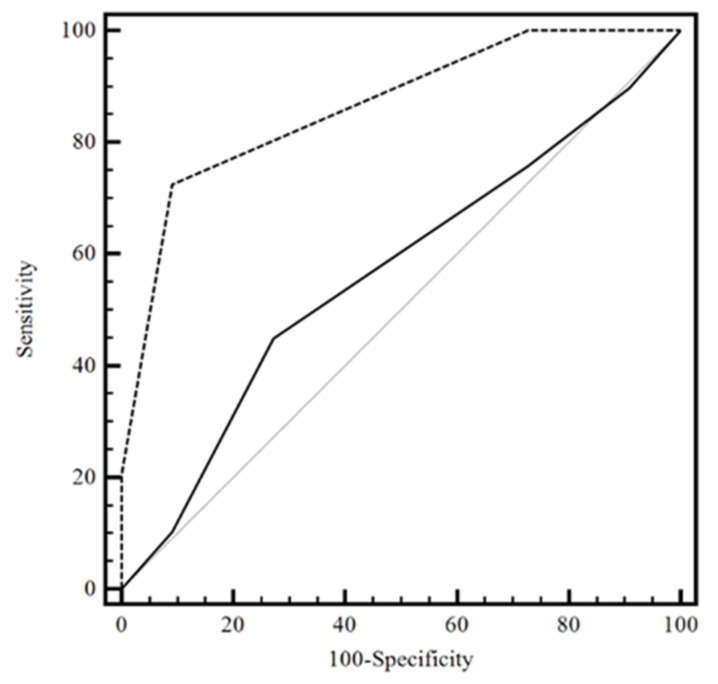
The considered ROC curve for clinical data prediction score with MMSE score and clinical data prediction without MMSE score. The value of the classification result produced best performance with an AUC value of 0.86 in clinical data prediction with MMSE score. Legend: −−−−clinical data prediction without MMSE score; ------- clinical data prediction with MMSE score; Grey solid line: clinical data prediction without MMSE score; ROC = Receiver Operating Characteristic; MMSE = Mini Mental State Examination; AUC = Area Under the Curve.

**Table 1 jpm-13-01386-t001:** MMSE scores grouped in 4 ranges of Cognitive Status.

MMSE Scores	CS
10–13	0
14–19	1
20–25	2
26–27	3
28–30	4

Legend: MMSE = Mini Mental State Examination; CS = Cognitive Status.

## Data Availability

Data are available upon request from the corresponding author.

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
