# Peer review of "Paving the Way for Predicting the Progression of Cognitive Decline: The Potential Role of Machine Learning Algorithms in the Clinical Management of Neurodegenerative Disorders"

_jpm, 2023, doi:10.3390/jpm13091386_

Round 1
Reviewer 1 Report

please refer to the attachment
Author Response
Dear Reviewer,
Thank you for your suggestions. We corrected all notes reported in the main text.
Best Regards
Desirèe Latella
Reviewer 2 Report
The paper topic is important but the current paper more suitable for conference 4 pages publication – to make this a journal paper, more figures,
Abstract
Need to give separate results for people with mild vs ppl with AD – also not clear from the abstract what is the prediction – the diagnosis (yes/no) or prognosis (likelihood of the MCI to become AD)?
Introduction
“ 47 million people were diagnosed with dementia” – not updated, from 9 bilioon, where you have over 500 milions over 80, at least half of them start suffering from MCI or AD
The literature review too short – need at least 20 papers
Methods – need to describe with figure, the data flow
No legend in the main figure
No clear separate between MCI and AD predictions
No discussion about prognosis of the disease
Author Response
Dear Reviewer,
Thank you for your suggestion. We corrected as following:
- Abstract
Need to give separate results for people with mild vs ppl with AD – also not clear from the abstract what is the prediction – the diagnosis (yes/no) or prognosis (likelihood of the MCI to become AD)?
Response: we created a flow chart to specify how we selected the sample (see figure 1). the aim of this study is to diagnosed the cognitive decline for MCI patients and AD patients. we added these in abstract section.
introduction
“ 47 million people were diagnosed with dementia” – not updated, from 9 bilioon, where you have over 500 milions over 80, at least half of them start suffering from MCI or AD
The literature review too short – need at least 20 papers
response: the literature was updated as you suggested.
-methods – need to describe with figure, the data flow
No legend in the main figure
No clear separate between MCI and AD predictions
No discussion about prognosis of the disease
response: we created the data flow. we added legend in the figure. the aim of this study is to diagnosed the cognitive decline for MCI patients and AD patients. we added these in abstract section.
best regards,
Desirèe Latella
Reviewer 3 Report
The nature of this research project is inherently tied to the quality of the data utilized for constructing and assessing models. Consequently, the dataset employed for predictive purposes necessitates thoroughly depicting various aspects, including the population, temporal context, environmental conditions, features, etc. Additionally, it is crucial to provide substantiated proof of the results' generalizability. A comprehensive discussion of results is also advisable. In the conclusion section, it is imperative to reframe and highlight the key findings of the research.
The overall quality of global English is acceptable, but some grammatical issues require attention.
Author Response
Dear reviewer,
thanks for your suggestions.
we increased the work with your suggestions. As regards the population used, a flow diagram was constructed to describe how we selected the population and we added the period in which the patients were recruited. the discussion has been implemented with a better description of the results as well as in the conclusions section.
Reviewer 4 Report
1. Mention your novelty of work in the introduction section in bullet form.
2. Discuss more about the findings of your work, why your work is more accurate as compare to the existing work.
3. Discuss more about your proposed model.
4. What is the significance of Geriatric Depression Scale in your article?
5. What is the significance of Cognitive Level in your article?
6. How your work is useful for the society? Please discuss it in conclusion section.
7. Mention the limitations of your proposed model.
Minor editing of English language required.
Author Response
- Mention the newness of your work in the introduction section as a bullet point.
we added a sentence in the introductory section.
2. Discuss more about the results of your work, why your work is more accurate than existing work.
I discussed the results better and made a comparison with another literature study about the novelty of our study.
3. Discuss more about the proposed model.
I have better discussed the model and the variables that we have considered in our work.
4. What is the significance of the Geriatric Depression Scale in your article?
it's a mistake I deleted the sentence.
5. What is the meaning of cognition level in your article?
by cognitive level we meant the state of the patient's cognitive profile. I changed Cognitive Level to Cognitive Status.
6. How is your work useful to society? Please discuss this in the conclusions section.
I added a sentence emphasizing this point in the conclusion section
7. Mention the limitations of the proposed model.
I added a paragraph about the limitations of the study.
we improved English.
best regards,
Desiree Latella
Round 2
Author Response
Dear reviewer,
thank you for your suggestions. We preferred to use MLM to indicate our approach.
thank you,
best regards,
Desiree Latella
Reviewer 2 Report
You improved the paper quality. Still for journal publication it is not sufficient - the analsyis, the number of results figures. There is no real compare to other similar methods, which make it difficult to compare and estimate the paper technical contribution
Author Response
Dear reviewer,
Thank you for your suggestions.
we added more informations about our model compared to literature to give more significativity to our results. we added in discussion section.
best regards,
Desiree Latella
Reviewer 4 Report
Authors have revised the article according to the review comments.